# Probiotic Properties and Safety Evaluation of *Lactobacillus plantarum* HY7718 with Superior Storage Stability Isolated from Fermented Squid

**DOI:** 10.3390/microorganisms11092254

**Published:** 2023-09-08

**Authors:** Hyeonji Kim, Myeong-Seok Yoo, Hyejin Jeon, Jae-Jung Shim, Woo-Jung Park, Joo-Yun Kim, Jung-Lyoul Lee

**Affiliations:** 1R&BD Center, hy Co., Ltd., 22, Giheungdanji-ro 24beon-gil, Giheung-gu, Yongin-si 17086, Republic of Korea; skyatk94@gmail.com (H.K.); audtmrl@hy.co.kr (M.-S.Y.); 10003012@hy.co.kr (H.J.); jjshim@hy.co.kr (J.-J.S.); 2Department of Marine Food Science and Technology, Gangneung-Wonju National University, Gangneung 25457, Republic of Korea; pwj0505@gwnu.ac.kr

**Keywords:** fermented foods, probiotics, *Lactobacillus plantarum*, storage stability, tight junction

## Abstract

The aim of this study was to identify new potential probiotics with improved storage stability and to evaluate their efficacy and safety. Sixty lactic acid bacteria strains were isolated from Korean traditional fermented foods, and their survival was tested under extreme conditions. *Lactobacillus plantarum* HY7718 (HY7718) showed the greatest stability during storage. HY7718 also showed a stable growth curve under industrial conditions. Whole genome sequencing revealed that the HY7718 genome comprises 3.26 Mbp, with 44.5% G + C content, and 3056 annotated Protein-coding DNA sequences (CDSs). HY7718 adhered to intestinal epithelial cells and was tolerant to gastric fluids. Additionally, HY7718 exhibited no hemolytic activity and was not resistant to antibiotics, confirming that it has probiotic properties and is safe for consumption. Additionally, we evaluated its effects on intestinal health using TNF-induced Caco-2 cells. HY7718 restored the expression of tight junction proteins such as zonular occludens (*ZO-1*, *ZO-2*), occludin (*OCLN*), and claudins (*CLDN1*, *CLDN4*), and regulated the expression of myosin light-chain kinase (*MLCK*), *Elk-1*, and nuclear factor kappa B subunit 1 (*NFKB1*). Moreover, HY7718 reduced the secretion of proinflammatory cytokines such as interleukin-6 (IL-6) and IL-8, as well as reducing the levels of peroxide-induced reactive oxygen species. In conclusion, HY7718 has probiotic properties, is safe, is stable under extreme storage conditions, and exerts positive effects on intestinal cells. These results suggest that *L. plantarum* HY7718 is a potential probiotic for use as a functional supplement in the food industry.

## 1. Introduction

Fermented foods and beverages are produced traditionally through growth and fermentation by microorganisms [1]. Fermented foods have received much attention worldwide [2]; indeed, a variety of studies show that fermented foods have health benefits, which include acting as antioxidants, reducing inflammation, and ameliorating diabetes, obesity, and metabolic syndrome [3,4]. Microbial activity can improve these functions further [5]; indeed, microorganisms regulate host health and metabolism by producing components such as signaling molecules, metabolites, and antimicrobial compounds [3,6].

The World Health Organization (WHO) defines probiotics as “Live microorganisms that when administered in adequate amounts, confer a health benefit on the host” [7]. Probiotics exert anti-inflammatory, antioxidant, and anti-obesity effects; they can also be beneficial for the skin [8,9,10,11] and modulate the intestinal microbial environment [12,13]. Probiotics in traditional fermented foods increase the nutritional status and potential health benefits of these foods [14]. *Lactobacillus* is a representative probiotic species present in fermented Korean traditional foods such as kimchi (fermented vegetables), jeotgal (fermented fish/seafood), and jang (fermented soy bean sauces) [15,16].

Probiotics are used widely by the food industry as functional ingredients and dietary supplements. The FAO/WHO suggest that to provide health benefits, probiotic products must maintain a cell viability of at least 10^6^ CFU (colony forming units)/g throughout their shelf life [17,18,19]. Thus, this must be considered during the production of dried probiotic preparations, functional food supplements, and fermented probiotic cultures. Dry probiotics (including freeze-dried products) are typically stable when stored at 4 °C; however, when added to products such as dietary supplements, powdered milk, and cereals, these dried probiotics must remain stable at higher temperatures. These products are generally sold at room temperature, and storing dried probiotics at unchilled temperatures can reduce transportation and storage costs [20]. Therefore, it is necessary to consider characteristics that maintain survival rates during storage in an ambient environment [21].

The purpose of this study was to screen *Lactobacillus* strains by conducting storage stability tests to identify those that have potential as a functional probiotic for use by the food industry. The survival rates of lactic acid bacteria (LAB) strains isolated from Korean traditional fermented foods were evaluated after storage under high-temperature and high-humidity conditions. The selected LAB strain, HY7718, was confirmed as a potential probiotic through whole genome sequencing, and by the analysis of probiotic properties and in vitro safety profiles. Finally, we conducted in vitro evaluations using intestinal cells to analyze its potential as a functional ingredient.

## 2. Materials and Methods

### 2.1. Sample Collection and Isolation of LAB Strains

LAB strains were obtained from various Korean traditional fermented foods at local traditional markets in Korea. Fermented food samples were collected in sterile containers and stored in a −80 °C deep freezer until used in the experiments. Solid samples were chopped finely and mixed with sterile phosphate buffered saline (PBS) at a ratio of 1:9 (*v*/*v*). The mixtures were then homogenized using a Polytron homogenizer (Kinematica, Malters, Switzerland). Liquid samples were vortexed vigorously. Prepared samples were serially diluted in 1/10 ratio with sterile PBS, spread onto MRS agar plates (BD Difco, Sparks, MD, USA), and incubated under anaerobic conditions for 48 h at 37 °C. Sixty cultured colonies were picked at random and streaked onto fresh MRS agar plates to obtain pure isolates. The list of picked bacterial strains is shown in Appendix A. The 60 bacterial strains were maintained at −80 °C as frozen glycerol (20% *v*/*v*) stocks. Bacteria were identified through 16S rRNA sequencing using universal primer set 27F/1492R (Macrogen, Seoul, Republic of Korea). The results of 16S rRNA sequencing were compared using the Basic Local Alignment Search Tool (BLAST) of the National Center for Biotechnology Institute (NCBI).

### 2.2. Probiotics Storage Stability Experiments

#### 2.2.1. Pre-Cultivation

The 60 strains isolated from fermented foods were inoculated into 10 mL of MRS broth in conical tubes and incubated under anaerobic conditions for 24 h at 37 °C. The cultures were then centrifuged at 4000× *g* for 20 min at 10 °C. The cell pellets were mixed thoroughly with 500 mL of sterilized 10% skim milk (BD Difco, Sparks, MD, USA) as a cryoprotectant. Finally, the cell suspensions were frozen at −80 °C for 6 h. Frozen samples were freeze-dried in a Freeze-dryer (Operon FDT-8620; Operon, Gimpo, Republic of Korea) from −80 °C to 20 °C over 72 h to evaluate storage stability.

#### 2.2.2. Storage Stability of Isolated LAB Strains

The lyophilized LAB strains were stored in a thermos-hygrostat at room temperature (25 °C). At this time, the LAB samples were unsealed to expose them to humidity of around 40%. The number of viable cells was counted once a week for 2 weeks. Stored samples were serially diluted in PBS and plated on MRS agar. The MRS agar plates were incubated anaerobically at 37 °C for 48 h and the number of colonies was counted. The survival rate of LAB strains during the storage period was calculated as follows:Survival Rate (%) = (Log CFU/g at specific time point)/(Log CFU/g at initial time point) × 100

#### 2.2.3. Lab-Scale Storage Test

The lab-scale storage stability of the selected strains was tested using a culture fermenter. LAB strains were pre-cultivated for 48 h at 37 °C in MRS broth under anaerobic conditions. Pre-cultured strains in MRS were inoculated into a lab-scale fermenter at a final concentration of 1% (*v*/*v*). The fermenter was run at 37 °C, with the pH maintained at 5.5 and stirring at 100 rpm, for 18 h. The pH during fermentation was adjusted using 15% ammonia solution. After fermentation, the culture broth was centrifuged at 4000× *g* for 20 min at 10 °C (Avanti JXN-26 centrifuge; Beckman Coulter (Brea, CA, USA)). The collected pellet was mixed thoroughly with an equal volume of 10% skim milk as a cryoprotectant. The suspensions were dispensed into Petri dishes and freeze-dried. The lyophilized strains were then stored in aluminum/PE pouches at 25 °C/40% humidity for 4 weeks. The number of CFU was measured four times (once per week over 4 weeks).

#### 2.2.4. Industrial Mass Production and Storage Stability of HY7718

HY7718 was cultured in a 2000 L plant-scale fermenter (CNS, Daejeon, Republic of Korea) to assess the possibility of industrial scale-up. The strain (total volume = 1200 L) was cultured at 37 °C (pH maintained at 5.5 with stirring at 100 rpm) for 18 h. The composition of the culture medium is shown in Table 1. Viable cell counts were measured by sampling every 3 h. Culture broth was centrifuged in an Alfa Laval BTPX centrifuge (Alfa Laval, Lund, Sweden) and the supernatant removed. The collected cells were mixed with an equal volume of sterile 10% skim milk as a cryoprotectant. The mixture was freeze-dried (from −80 °C to 20 °C) using a Tofflon Lyophilizer (LYO 15; Tofflon, Shanghai, China) over 72 h. The lyophilized powder was sealed and stored in aluminum/PE pouches at 25 °C/40% humidity. The number of HY7718 cells was counted four times (once per week for 4 weeks).

### 2.3. Bacterial Culture and Sample Preparation for In Vitro Assays

For in vitro assays, LAB strains were cultured anaerobically in MRS broth for 24 h at 37 °C and then centrifuged at 4000× *g* for 20 min. The harvested cells were washed twice and resuspended in sterile PBS. Type strain *Lactobacillus plantarum* ATCC 14917 (ATCC 14917) was used as a LAB control to compare probiotic characteristics and in vitro effects with those of the selected strain HY7718.

### 2.4. Whole Genome Sequencing of Lactobacillus plantarum HY7718

Genomic DNA (gDNA) was extracted from HY7718 using the Qiagen MagAttract HMW DNA Kit (Qiagen, Hilden, Germany). The concentration of extracted gDNA was quantified using the Qubit 2.0 fluorometer (Invitrogen, Carlsbad, CA, USA). The whole genome of *Lactobacillus plantarum* HY7718 was sequenced on the Illumina MiSeq (Illumina, Inc., San Diego, CA, USA) and PacBio RS II (Pacific Biosciences Inc., Menlo Park, CA, USA) platforms at CJ bioscience (Seoul, Republic of Korea). Library construction was performed using the TruSeq DNA Library LT Kit for Illumina (Illumina, San Diego, CA, USA) and the SMRTbell Template Preparation Kit for PacBio (100-938-900). The resulting reads were assembled by Unicycler v 0.4.3 using quality controlled Illumina MiSeq and PacBio read data. Gene prediction and annotation were performed using EzBioCloud database at CJ bioscience. Protein-coding DNA sequences (CDSs) were predicted by Prodigal 2.6.2. Then, the tRNA and rRNA sequences of HY7718 were searched for using tRNAscan-SE 1.3.1 and Rfam 12.0 databases, respectively. After gene prediction, Cluster of Orthologous Groups (COG) annotation was performed using EggNOG 4.5 (http://eggnogdb.embl.de (accessed on 15 September 2022)). Virulence factors were investigated using the Virulence factors database (VFDB; http://www.mgc.ac.cn/VFs/ (accessed on 15 September 2022)). Antibiotic resistance genes were searched for using The Comprehensive Antibiotics Resistance Database (CARD; https://card.mcmaster.ca/ (accessed on 15 September 2022)).

### 2.5. Probiotic Characterization 

#### 2.5.1. Caco-2 Cell Adhesion Assay

Adhesion of HY7718 and ATCC 14917 to intestinal Caco-2 cells was measured as described previously [22]. Caco-2 cells were cultured at 37 °C/5% CO_2_ in Minimum Essential Medium (MEM, Welgene, Daegu, Republic of Korea) supplemented with 10% heat-inactivated fecal bovine serum (FBS, Gibco, Waltham, MA, USA) and 1% antibiotic-antimycotic (Gibco, Waltham, MA, USA). Caco-2 cells were seeded into 6-well plates at a density of 2.0 × 10^5^ cells/well and incubated until they reached 100% confluence. The medium was removed, the cells were washed twice with sterile PBS, and the medium was replaced with MEM without FBS and antibiotic-antimycotic. Cultured HY7718 and ATCC 14917 were diluted in PBS and added to each well at a concentration of 1 × 10^8^ CFU/mL. Plates were incubated for 2 h at 37 °C/5% CO_2_. After the bacteria had adhered to Caco-2 cells, the cells were washed four times with sterile PBS and treated with 0.025% Trypsin-EDTA (Gibco, Waltham, MA, USA) to remove the cells from the plate. The collected cells and attached bacteria were serially diluted in 1/10 ratio with sterile PBS, spread onto MRS agar plates, and incubated for 48 h at 37 °C. Colonies were then counted.

#### 2.5.2. Survival Rates in the Simulated Gastro Intestinal Tract 

Simulated Gastro Intestinal Tract (GIT) tests were performed to evaluate survival of HY7718 and ATCC 14917 under physiological conditions. In vitro digestion assays were performed to measure tolerance to saliva (pH 7), gastric fluid (pH 3), and bile salts (pH 7), as described by Mo et al. [23]. Simulated salivary fluid (SSF), simulated gastric fluid (SGF) and simulated intestinal fluid (SIF) were pre-made and pre-heated to 37 °C before the simulated GIT tests. Briefly, 5 mL cultures of HY7718 and ATCC 14917 were prepared in a 50 mL conical tube. The oral phase was conducted by mixing 4 mL of α-amylase (6.55 mg/mL) and 26 μL of 0.3 M CaCl_2_ to make an SSF solution. Next, 1 M NaOH was added to adjust the pH to 7.0 and the mixture was incubated at 37 °C for 2 min. The gastric phase was performed by mixing 694 μL of distilled water (D.W.), 9.1 mL of pepsin (0.07 mg/mL), and 6 μL of 0.3 M CaCl_2_ to make an SGF solution. Next, 1 M HCl was added to adjust the pH to 3.0 and the mixture was incubated at 37 °C under shaking for 2 h. The intestinal phase was performed by mixing 1.31 mL of D.W., 2.5 mL of 160 mM bile extract (Sigma-Aldrich, St Louis, MO, USA), 16 mL of pancreatic solution (22.15 mg/mL), and 40 μL of 0.3 M CaCl_2_ to make an SIF solution. Next, 1 M NaOH was added to raise the pH to 7.0 and the mixture was incubated at 37 °C for 2 h with shaking. After each phase, aliquots of the reaction mixture were obtained, and viable bacteria were counted to measure the survival rates:Survival Rate (%) = (CFU/g at specific time point)/(CFU/g at initial time point) × 100

#### 2.5.3. Safety Assessment 

The hemolytic activity of HY7718 was tested as recommended by the American Society for Microbiology. The strains were cultured for 18 h at 37 °C in MRS broth (BD Difco, Sparks, MD, USA). The cultured LAB strains were then streaked in blood agar plate supplemented with 5% sheep blood (KisanBio, Seoul, Republic of Korea). After incubating the strains at 37 °C for 48 h, hemolytic activity was confirmed.

The minimum inhibitory concentrations (MIC) of nine antibiotics (ampicillin, vancomycin, gentamicin, kanamycin, streptomycin, erythromycin, clindamycin, tetracycline, and chloramphenicol) were measured using MIC Test Strips (Liofilchem, Via Scozia, Roseto degli Abruzzi, Italy) in accordance with European Food Safety Authority (EFSA) guidelines. Sterilized cotton swabs were soaked in bacterial cultures and streaked onto MRS agar plates to form a lawn. MIC Test Strips were placed on the agar plates and incubated at 37 °C for 48 h. The lower value of the strip around which the bacteria did not grow was measured as the MIC.

### 2.6. Cell Culture and Sample Treatments

The Caco-2 human colon epithelial cell line was purchased from the American Type Culture Collection (Manassas, VA, USA) and maintained at 37 °C/5% CO_2_ in MEM containing 10% heat-inactivated FBS and 1% antibiotic-antimycotic. Cells were seeded into 12-well plates at a density of 1.0 × 10^5^ cells/well, and stabilized for 24 h. Cells were allowed to differentiate for 21 days, during which the medium was replaced every 2–3 days. After differentiation, the growth medium was removed and replaced with MEM without FBS and antibiotics. Next, 1.0 × 10^7^ CFU/well of HY7718 and ATCC 14917 were added to the wells (1.0 × 10^7^ CFU/1.0 × 10^5^ cells) and incubated for 2 h. After that, 100 ng/mL of tumor necrosis factor-α (TNF, R&D Systems, Minneapolis, MN, USA) was added to the cells for 24 h.

### 2.7. Isolation of Total RNA, cDNA Synthesis, and Real-Time PCR 

Total RNA was isolated from TNF-induced Caco-2 cells using the easy-spin Total RNA Extraction Kit (iNtRON Biotechnology, Seoul, Republic of Korea). Next, cDNA was synthesized at 37 °C for 60 min using the Omniscript Reverse Transcription Kit (Qiagen, Hilden, Germany). The cDNA samples were analyzed using the QuantStudio 6 Flex Real-time PCR System (Applied Biosystems, Foster City, CA, USA). Real-time PCR was performed using Gene Expression Master Mix (Applied Biosystems) and mouse-specific TaqMan probes. The genes analyzed in this study were as follows: Glyceraldehyde-3-phosphate dehydrogenase (*GAPDH*; Hs99999915_g1), tight junction protein 1 (*ZO-1*; Hs01551871_m1), tight junction protein 2 (*ZO-2*; Hs00910543_m1), occludin (*OCLN*; Hs05465837_g1), claudin-1 (*CLDN1*; Hs0021623_m1), claudin-4 (*CLDN4*; Hs00976831_s1), myosin light-chain kinase (*MLCK*; Hs00364926_m1), ELK1 (*Elk-1*; Hs00901847_m1), nuclear factor kappa B subunit (*NFKB1*; Hs00765730_m1), mucin 2 (*MUC2*; Hs03005103_g1), and mucin4 (*MUC4*; Hs00366414_m1). *GAPDH* gene was used for normalization of other gene expression data.

### 2.8. Measurement of Cytokine Secretion

Secretion of IL-6 and IL-8 into Caco-2 cell culture medium was measured using the BD OptEIA™ Human IL-6 ELISA Set and the Human IL-8 ELISA Set (BD Biosciences, San Diego, CA, USA; BD 555220 and BD 55244). Absorbance at 450 nm was measured in a BioTek^®^ Synergy HT Microplate reader (Santa Clara, CA, USA).

### 2.9. Evaluation of Intracellular Reactive Oxygen Species (ROS) Production 

Caco-2 cells were seeded (1.0 × 10^4^ cells/well) into 96-well plates and allowed to stabilize. The cells were then pre-treated for 24 h with 1.0 × 10^6^ CFU/well of HY7718 and ATCC 14917 (1.0 × 10^6^ CFU/1.0 × 10^4^ cells). The culture medium was removed and 100 μL of H_2_O_2_ was added for 1 h. Next, Caco-2 cells were stained for 10 min with 5 μM DCF-DA reagent. DCF-DA-stained cells were washed twice with PBS and analyzed under a Zeiss Axiovert 200 M microscope (Carl Zeiss AG, Thornwood, NY, USA). Fluorescence at 485 nm (excitation)/535 nm (emission) was measured by a BioTek^®^ Synergy HT Microplate reader (Santa Clara, CA, USA) to evaluate ROS production.

### 2.10. Statistical Analysis

All data are presented as the mean ± standard deviation (SD). With respect to the acid tolerance and survival rates in the GIT, differences between HY7718 and ATCC 14917 were analyzed using an unpaired Student’s *t*-test. For in vitro assay data, differences between groups were analyzed by one-way ANOVA followed by Tukey’s post-hoc test. Statistical analysis was performed using GraphPad Prism 10 (GraphPad Software, San Diego, CA, USA), and *p* < 0.05 was considered significant.

## 3. Results

### 3.1. Storage Stability of Isolated LAB Strains

#### 3.1.1. Storage Stability of LAB Strains Isolated from Korean Fermented Foods

In this study, we isolated a total of 60 strains of LAB from Korean traditional fermented foods (*Lactobacillus plantarun*: 20 strains, *Lactobacillus paracasei*: 40 strains). To identify strains with good storage stability, we measured the survival of bacterial strains at a high temperature (25 °C) and high humidity (40%). As shown in Figure 1A, 20 strains showed a survival rate of 60% or more after 1 week of storage. Among these, six showed survival rates of ≥90% (*L. plantarum* #7, #8, #16, #18, #19, #20) and five strains showed survival rates of 80–89% (*L. plantarum* #2, #4, #10, #13, #15). Other strains measured lower than 1 × 10^7^ CFU, so were labeled Low detection (L.D.) after 1 week. Only eight strains showed survival rates of ≥80% by week 2 (#4, #7, #8, #15, #16, #18, #19, and #20). Thus, six LAB strains (*L. plantarum* #7, #8, #16, #18, #19, and #20) showed high storage stability under harsh conditions during Weeks 1 and 2.

#### 3.1.2. Lab-Scale Stability of the Screened LAB Strains 

Next, the stability of the six selected strains was evaluated when grown in Lab-scale fermenter at 25 °C and 40% humidity. The LAB strains were cultured under these harsh conditions for 1 month, and the number of viable bacteria was measured weekly. Figure 1B shows that all strains survived during the first week. Survival cells of *L. plantarum* #7, #8, and #16 decreased gradually in the second, third, and fourth weeks. The number of viable cells of #18 and #20 was similar until the third week, but survival of #20 was higher than that of #18 by the fourth week. *L. plantarum* #19 was most stable of the six strains under high-temperature/high-humidity conditions. Finally, we selected *L. plantarum* #19 and renamed the strain *Lactobacillus plantarum* HY7718 (HY7718).

#### 3.1.3. Industrial Culture and Storage Stability Testing of HY7718

To confirm that HY7718 is suitable for commercial use, we performed a large-scale culture in an industrial fermenter. Figure 1C shows that the growth curve of HY7718 reached the stationary phase after 18 h in a 1200 L fermentation volume. After culture and concentration, lyophilized HY7718 was stored for 4 weeks under the same conditions used for the laboratory experiments. As shown in Figure 1D, the HY7718 (Log) counts were 11.98, 11.93, 11.69, 11.54, and 11.20 at the first, second, third, and fourth weeks, respectively. In other words, the number of HY7718 cells remained constant at 25 °C/40% humidity over the 4 weeks.

### 3.2. Whole-Genome Sequencing of HY7718

The complete genome of *Lactobacillus plantarum* HY7718 consists of a circular chromosome of 3,262,430 bp, with a guanine + cytosine (G + C) content of 44.5%. HY7718 was predicted to contain 3056 protein CDSs, 68 tRNA-encoding genes, and 16 rRNA-encoding genes (Table 2). The CDSs were classified into COG functional categories (Appendix A). In addition, the sequencing of HY7718 identified no virulence factors or antibiotic resistance genes. The genome map of HY7718 is presented in Figure 2.

### 3.3. Probiotic Characterization of HY7718

#### 3.3.1. Adhesion of HY7718 to Intestinal Cells

The ability of bacteria to adhere to the intestinal epithelium is a major precondition for the colonization of the digestive tract. To assess the ability of HY7718 to adhere to intestinal cells, we conducted an assay using Caco-2 cells. Figure 3A shows that 20.63 ± 0.88% and 3.34 ± 0.41%, respectively, of HY7718 and ATCC 14917 bacteria adhered to CaCo-2 cells (*p* < 0.001). These data suggest that HY7718 is able to colonize the digestive tract.

#### 3.3.2. Survival of HY7718 in the Simulated GIT

Potential probiotic strains must be able to tolerate gastric juices and bile secretions if they are to pass through and colonize the intestine [24]. Therefore, we used the simulated GIT to test the resistance of HY7718 to highly acidic gastric secretions and bile juices. As shown in Figure 3B, there was no significant change in the survival rate in SSF of both HY7718 and type strain ATCC 14917. On the other hand, in SGF, the survival rate of HY7718 was significantly higher than the type strain. In SIF, HY7718 showed a higher survival rate than the type strain, but it was not statistically significant. Thus, it was confirmed that HY7718 has high viability in the gastric and intestinal environment, especially in the gastric conditions.

#### 3.3.3. Safety Assessment

To ensure that HY7718 is safe for human consumption, we examined its hemolytic activity and antibiotic resistance. As shown in Figure 3C, HY7718 did not show hemolytic activity. Next, we measured the MIC cut-off values of nine antibiotics using MIC Test Strips. Table 3 shows the MIC values of antibiotics for HY7718; this strain was not resistant to any of the antibiotics tested. Thus, HY7718 has potential for use as a probiotic.

### 3.4. Effects of HY7718 on Expression of Tight-Junction (TJ)-Related Genes by TNF-Treated Caco-2 Cells 

Next, we examined the effects of HY7718 on the expression of genes related to TJ formation in a TNF-induced Caco-2 cell model. TNFα disrupts the intestinal epithelial TJ barrier, resulting in increased intestinal permeability [25]. The mRNA expression of *ZO-1*, *OCLN*, *CLDN1*, and *CLDN4* was decreased by TNF-α and significantly recovered upon HY7718 treatment. In contrast, treatment with ATCC 14917 did not significantly affect the mRNA levels of these genes (Figure 4A,C–E). *ZO-2* mRNA expression was decreased by TNF-α and increased by HY7718 treatment, but there was no significance between the groups (Figure 4B). These results indicate that HY7718 can restore intestinal epithelial integrity by regulating tight-junction-related genes. 

### 3.5. Effects of HY7718 Treatment on Myosin Light-Chain Kinase (MLCK) Pathway Gene Expression in TNF-Induced Caco-2 Cells 

MLCK plays a role in regulating TJ-associated intestinal permeability through TNF-α-induced NF-κB activation [26]. Therefore, we examined the expression of genes related to the MLCK pathway (i.e., *MLCK*, *Elk-1*, and *NFKB1*). The expression of *MLCK*, *Elk-1*, and *NFKB1* was significantly increased by approximately 1.5-fold by TNF-α stimulation and adjusted to normal levels by HY7718 treatment. By contrast, the expression levels of all three genes in ATCC 14917-treated cells tended to be lower than in TNF-induced cells, but the differences were not significant (Figure 5A–C). In particular, the mRNA expression of *MLCK* and *NFKB1* was significantly higher in cells treated with HY7718 than those treated with ATCC 14917 (*p* < 0.001). These results indicate that HY7718 regulates the MLCK pathway to maintain the integrity of intestinal epithelial TJs.

### 3.6. Effect of HY7718 on Secretion of Inflammatory Cytokines by TNF-Induced Caco-2 Cells

Next, we examined the secretion of proinflammatory cytokines IL-6 and IL-8 in TNF-treated Caco-2 cells. As shown in Figure 6, the secretion of IL-6 and IL-8 by TNF-treated cells was significantly higher than that by untreated cells. The secretion of IL-6 reduced significantly after HY7718 and ATCC 14917 treatment. On the other hand, the secretion of IL-8 decreased in HY7718 treatment, but not in ATCC 14917 treatment. In addition, it was confirmed that HY7718 significantly reduced the secretion of IL-6 and IL-8 compared to ATCC 14917. These results indicate that HY7718 can inhibit the secretion of inflammatory cytokines from intestinal epithelial cells more effectively than type strains.

### 3.7. Effects of HY7718 on ROS Production by H_2_O_2_-Treated Cells

ROS produced by cells subjected to oxidative stress is a major cause of intestinal barrier dysfunction [27]. We found that the production of intracellular ROS by cells induced by H_2_O_2_ increased significantly to 211.6% (*p* < 0.001) of that by untreated cells (Figure 7A,B); however, HY7718 treatment led to a significant reduction in ROS levels to 147.6% (*p* < 0.001). ATCC 14917 slightly reduced the ROS levels to 190.5%, but it was not significant.

## 4. Discussion

Probiotics are promoted as functional foods by the global food industry. For commercial use, probiotics need to be stable, safe, and show good probiotic functions when added to foods [21].

Here, we isolated and evaluated a new probiotic named *Lactobacillus plantarum* HY7718. HY7718, which was derived from Korean traditional fermented foods, showed high stability during storage under extreme conditions and in lab-scale fermenter cultures. Furthermore, we used a large-scale fermenter to confirm the potential of HY7718 for industrial use. A bacterial growth curve generally comprises four phases: lag phase, exponential phase, stationary phase, and death phase [28]. The growth curve of HY7718 could be divided into three distinct phases: 0–3 h, lag phase; 3–9 h, exponential phase; and 9–18 h, stationary phase. In other words, HY7718 showed a normal growth curve up to the stationary phase in large-scale culture, and did not reach the death phase for 18 h. In addition, HY7718 produced at plant-scale showed high storage stability. This strongly suggests that HY7718 is an industrially useful strain suitable for mass production.

Whole genome sequencing of potential probiotics is required to analyze genetic data related to metabolic activity, virulence factors, and antibiotic resistance [29]. The DNA sequence of HY7718 is 3.26 Mbp in length, with a G + C content of 44.5%, and 3056 CDSs. We also confirmed that HY7718 does not predict virulence factors or antibiotic resistance genes. These results suggest that HY7718 will be safe for human consumption.

We examined the safety profile of HY7718 further in in vitro tests. Hemolysis and antibiotic resistance are potentially harmful to both the human host and associated intestinal flora [30,31]. Therefore, the safety of probiotics must be proven prior to consumer consumption. Here, we found that HY7718 exhibited no hemolytic activity. In addition, the strain was not resistant to antibiotics, as assessed according to EFSA guidelines. Therefore, strain HY7718 appears to be safe for use as a probiotic.

The health benefits of probiotics can only be realized if the bacteria survive the harsh conditions in the GIT [32]. Therefore, we evaluated the survival of HY7718 in a simulated GIT model. HY7718 was more stable than type strain ATCC 14917 under simulated GIT conditions. The ability to adhere to the intestinal mucus layer, and to epithelial cells, is also critical for the bacterial colonization of the digestive tract [33]. The adhesion of HY7718 to CaCo-2 cells was significantly greater than that of ATCC 14917.

*Lactobacillus* species improve the intestinal barrier function by enhancing TJ formation [34,35]. TJs are multifunctional complexes that maintain the intestinal barrier and regulate the passage of ions, water, essential nutrients, and other molecules [36,37]. Many TJ proteins are transmembrane proteins such as zonular occludens (ZO-1, ZO-2), occludin, and claudins [38]. We found that HY7718 significantly restored the expression of *ZO-1*, *ZO-2*, *OCLN*, *CLDN1*, and *CLDN4* mRNA by TNF-treated epithelial cells. These results suggested that HY7718 has the potential to improve intestinal permeability by restoring TJs. In other words, HY7718 may improve a leaky gut.

TNFα-induced TJ permeability is mediated by the activation of the ETS transcription factor Elk-1, followed by the activation of MLCK promoter activity and gene transcription [39]. MLCK activation via phosphorylated MLC increases TJ permeability and disrupts structural proteins ZO-1 and occludin [40,41]. In addition, NF-κB signaling regulates intestinal TJs by upregulating MLCK [42]. Here, we found that the elevated expression of *MLCK*, *Elk-1*, and *NFKB1* mRNA by TNFα was inhibited by HY7718. This suggests that HY7718 increases the expression of TJ proteins by modulating the MLCK and NF-κB signaling pathways.

Previous studies show that proinflammatory cytokines such as IL-6 promote dysfunction of epithelial TJs and increase intestinal permeability [43,44]. In addition, IL-8 regulates intestinal permeability by downregulating TJ proteins [45]. Our ELISA data show that the secretion of IL-6 and IL-8 by TNFα-induced Caco-2 cells was inhibited significantly by HY7718. In addition, it inhibited cytokine secretion to a greater extent than type strain ATCC 14917. Thus, HY7718 prevents intestinal barrier dysfunction by inhibiting the secretion of proinflammatory cytokines.

Finally, oxidative stress induced by ROS disrupts TJs. Excessive ROS can cause oxidative stress and increase epithelial permeability [46]. We confirmed that HY7718 reduced the amount of H_2_O_2_-induced ROS produced by Caco-2 cells. ROS also induce the production of cytokines, thereby increasing GIT inflammation in conditions such as IBD and CD (Crohn’s disease) [47]. The overexpression of NF-κB induced by ROS increases the transcription of genes encoding proinflammatory cytokines such as IL-6 and IL-8, thereby increasing intestinal permeability [48]. Taken together, our data suggest that HY7718 maintains TJs, regulates the production of inflammatory cytokines, and prevents oxidative stress.

## 5. Conclusions

*L. plantarum* HY7718, isolated from Korean traditional fermented foods, shows strong storage stability at high temperatures and high humidities. The bacterium shows good probiotic properties and safety. In addition, the strain shows robust growth, with the potential for industrial scale-up. Furthermore, in vitro experiments using Caco-2 cells confirmed that HY7718 is better than the type strain at suppressing inflammatory responses and maintaining TJs in the intestinal epithelium. Thus, we propose HY7718 as a new probiotic strain that will be of use to the food industry.

## Figures and Tables

**Figure 1 microorganisms-11-02254-f001:**
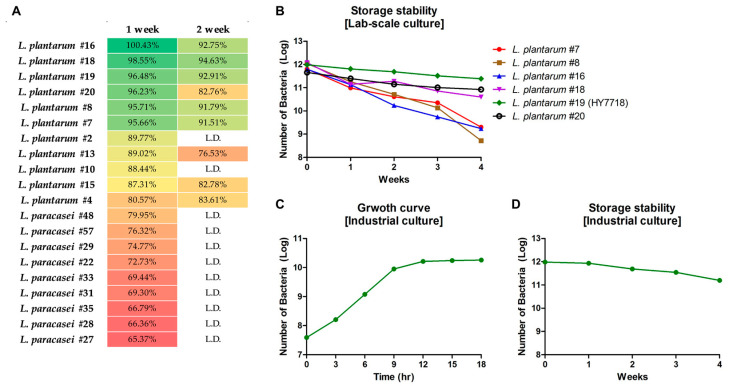
Storage stability tests. (**A**) Screening of bacterial strains isolated from Korean traditional fermented foods. (**B**) Survival of screened six bacterial strains (Log) grown in a lab-scale fermenter. (**C**) Growth curve of HY7718 in industrial tests. (**D**) Survival of HY7718 (Log) in industrial tests. L.D., Low detection (Week 1, <1 × 10^7^ CFU; Week2, <1 × 10^5^ CFU).

**Figure 2 microorganisms-11-02254-f002:**
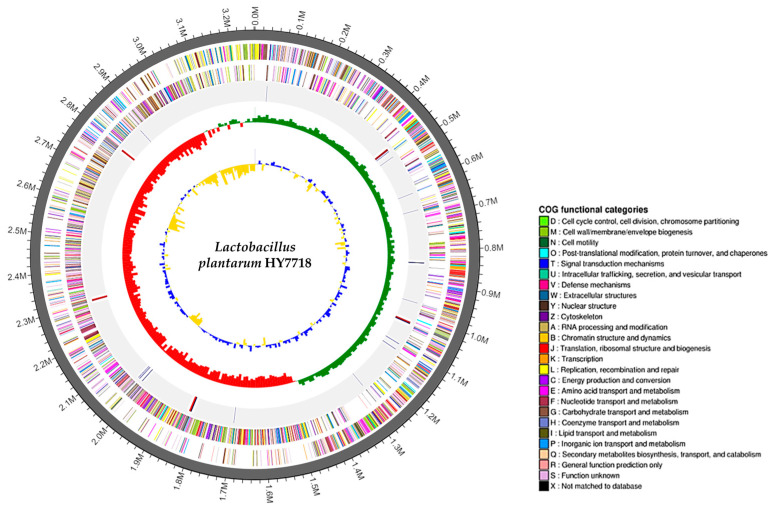
Genome map of *Lactobacillus plantarum* strain HY7718. Genome size (first outer circle). The Forward strand CDSs (second circle) and Reverse strand CDSs (third circle) are displayed in different colors according to the COG classification. Blue lines indicate tRNAs and red lines indicate rRNAs (fourth circle). Green and red peaks indicate the GC skew (fifth circle). Blue and yellow peaks indicate the GC ratio (inner circle). G: Guanine, C: Cytosine, CDS: Protein-coding DNA sequences, COG: Clusters of Orthologous Groups.

**Figure 3 microorganisms-11-02254-f003:**
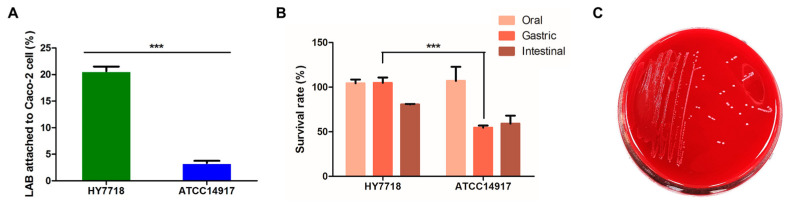
Probiotic characterization of HY7718. (**A**) Adhesion ability, (**B**) survival rate in simulated GIT, and (**C**) hemolytic activity. Results are presented as the mean ± SD. Significant differences are indicated as *** *p* < 0.001, compared with Type strain ATCC 14917. HY7718, *Lactobacillus plantarum* HY7718; ATCC 14917, *Lactobacillus plantarum* ATCC 14917.

**Figure 4 microorganisms-11-02254-f004:**
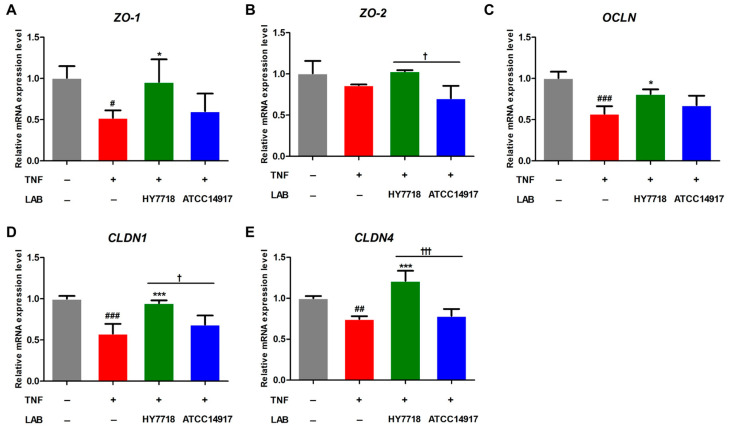
Effect of HY7718 on expression of tight-junction-related genes by TNF-induced Caco-2 cells. Levels of (**A**) *ZO*-1, (**B**) *ZO*-*2*, (**C**) *OCLN*, (**D**) *CLDN1*, and (**E**) *CLDN4* mRNA. Significant differences are indicated as ^#^
*p* < 0.05, ^##^
*p* < 0.01, and ^###^
*p* < 0.001 (untreated vs. TNF-treated); * *p* < 0.05 and *** *p* < 0.001 for TNF vs. probiotics, ^†^
*p* < 0.05 and ^†††^
*p* < 0.001 for HY7718 vs. Type strain ATCC 14917. LAB: Lactic acid bacteria, TNF: TNFα, HY7718: *Lactobacillus plantarum* HY7718, ATCC 14917: *Lactobacillus plantarum* ATCC 14917, ZO-1: Zonular occluden-1, ZO-2: Zonular occluden-2, OCLN: Occludin, CLDN1: Claudin-1, CLDN4: Claudin-4.

**Figure 5 microorganisms-11-02254-f005:**
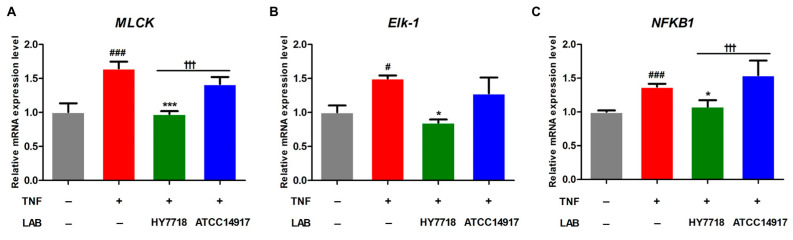
Effect of HY7718 on expression of MLCK pathway-related genes by TNF-induced Caco-2 cells. Levels of (**A**) *MLCK*, (**B**) *Elk*-*1*, and (**C**) *NFKB1* mRNA. Results are presented as the mean ± SD. Significant differences are indicated as ^#^
*p* < 0.05 and ^###^
*p* < 0.001 for untreated vs. TNF-treated cells, * *p* < 0.05 and *** *p* < 0.001 for TNF vs. probiotics, and ^†††^
*p* < 0.001 for HY7718 vs. Type strain ATCC 14917. LAB: Lactic acid bacteria, TNF: TNFα, HY7718: *Lactobacillus plantarum* HY7718, ATCC 14917: *Lactobacillus plantarum* ATCC 14917, MLCK: Myosin light-chain kinase, Elk-1: ETS like-1, NFKB1: Nuclear factor kappa subunit.

**Figure 6 microorganisms-11-02254-f006:**
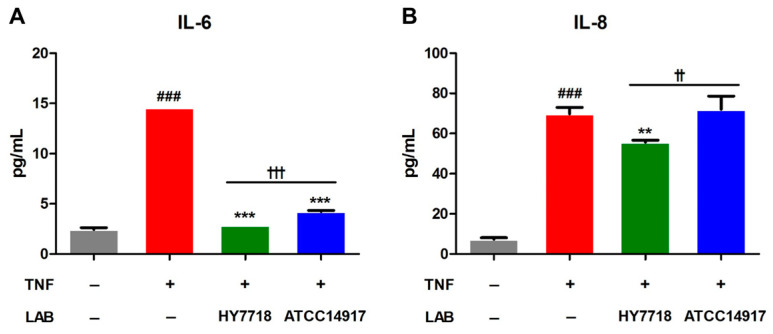
Effect of HY7718 on cytokines secretion by TNF-induced Caco-2 cells. The concentration of (**A**) IL-6 and (**B**) IL-8. Results are presented as the mean ± SD. Significant differences are indicated as ^###^
*p* < 0.001 for untreated vs. TNF-treated cells, ** *p* < 0.01 and *** *p* < 0.001 for TNF vs. probiotics, and ^††^
*p* < 0.01 and ^†††^
*p* < 0.001 for HY7718 vs. Type strain ATCC 14917. LAB: Lactic acid bacteria, TNF: TNFα, HY7718: *Lactobacillus plantarum* HY7718, ATCC 14917: *Lactobacillus plantarum* ATCC 14917, IL-6: Interleukin-6, IL-8: Interleukin-8.

**Figure 7 microorganisms-11-02254-f007:**
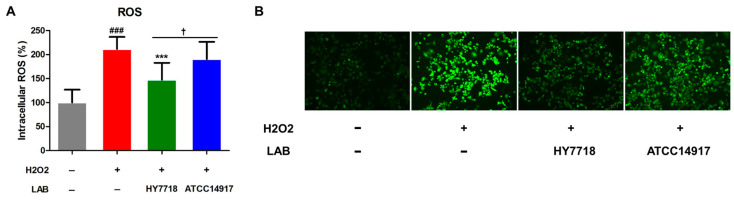
Antioxidant effects of HY7718 in H_2_O_2_-treated Caco-2 cells. Results are presented as the mean ± SD. Significant differences are indicated as ^###^
*p* < 0.001 for untreated vs. TNF-treated cells, *** *p* < 0.001 for TNF vs. probiotics, and ^†^
*p* < 0.05 for HY7718 vs. Type strain ATCC 14917. ROS: Reactive oxygen species, LAB: Lactic acid bacteria, H_2_O_2_: Hydrogen peroxide, HY7718: *Lactobacillus plantarum* HY7718, ATCC 14917: *Lactobacillus plantarum* ATCC 14917.

**Table 1 microorganisms-11-02254-t001:** Composition of the culture medium in the industrial fermenter.

Ingredient	Concentration (g/L)
Glucose	50
Yeast Extract	45
Soy peptone	25
K_2_HPO_4_	2
L-ascorbate	1
Tween 80	1
MgSO_4_	0.25
MnSO_4_	0.25
FeSO_4_	0.01

**Table 2 microorganisms-11-02254-t002:** Genomic information for *Lactobacillus plantarum* HY7718.

Features	Terms
Sequencing platforms	Illumina MiSeq PacBio RS II
Libraries used	TruSeq DNA Library LT Kit SMRTbell^®^ Prep Kit
Genome size (bp)	3,262,430
GC content (%)	44.5
Coding genes (CDSs)	3056
tRNA genes	68
rRNA genes	16

**Table 3 microorganisms-11-02254-t003:** MIC cut-off values of strain HY7718.

Antibiotic	EFSA	HY7718
*Ampicillin*	2	0.032
*Vancomycin*	n.r. ^1^	n.r.
*Gentamycin*	16	12
*Kanamycin*	64	64
*Streptomycin*	n.r.	n.r.
*Erythromycin*	1	0.5
*Clindamycin*	2	0.032
*Tetracyclin*	32	12
*Chloramphenicol*	8	6

^1^ n.r.; not required.

## Data Availability

The data presented in this study are available in the article and Appendix A.

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
