# Peer review of "Probiotic Properties and Safety Evaluation of Lactobacillus plantarum HY7718 with Superior Storage Stability Isolated from Fermented Squid"

_microorganisms, 2023, doi:10.3390/microorganisms11092254_

Round 1

Reviewer 1 Report

This paper is the result of confirming the effect of only the HY7718 strain using Caco2 cells, so conclusions and discussions limited to this should be made. In other words, the effect of interaction with other strains in the intestine when ingesting the HY7718 strain as a probiotic has not been verified, so conclusions or discussions should only explain the cell line experiment results and conclusions. Since this thesis is the result of the effect of HY7718 on Caco 2 cells, it should also be explained that there may be other results when ingested as probiotics.

For example, the following should be modified:

Line 428-430. According to previous studies, Lactobacillus improves gastrointestinal symptoms, including irritable bowel syndrome (IBD), as well as the composition of the intestinal flora [34-36].

Since the change in flora was not measured in the paper, it would be better to delete it.

Line 436-437. It is unreasonable to conclude that intestinal leakage can be improved because TJ integrity is improved only as a result of Caco2 cells. Interactions between strain HY7718 and other organisms in the intestine should be considered.

When selecting probiotics strains, stability is evaluated after acid resistance and bile resistance are mainly evaluated. Do you have evaluation results for acid resistance and bile resistance?

Line 470-471. Since this experiment is the result of an in vitro experiment, please delete this part.

Reviewer 2 Report

Comments to the authors

The authors present an interesting in vitro investigation on the potential characteristics of Lactobacillus plantarum HY7718, isolate obtained by fermentation of squid (traditional and/or local type product) throught initial screening of several isolates that document probiotic properties such as resistance to gastrointestinal trnasit under certain adverse conditions:saliva, acidi conditions (gastric fluid) and resistance to bile salts. in addition, its adhesion capacity, supported by a Caco-2 epitelial cells model in a comparative study with the reference strain of L. plantarum (ATCC14917).

The autthors also analyze the stability and safety of the product (hemolity activity) with posssibilites at the industrial level together with the role of this L. plantarum HY7718 isolate on a pro-inflammatory microenviroment induced by stimulated of Caco-2 cells with TNF- alpha, evaluating the expression of molecules related with resistance of the gastrointestinal tract barrier such as ZO binding proteins, claudin, mucins and pathaway of signalling NF-kB by real-time PCR. Also, the release of twoimportant pro-inflammatory cytokine (IL-6, IL-8) and the intracelllular production of ROS, and the genomic characterization of these isolate. 

Althought the authors results show that this strain of L.platarum HY7718 has probitic potential and superior storage stabilit, it is important to to reinforce certain points in the presentation of your manuscript:

1. In the materials and methods section, page 4, line ·#131, decribes the control used in this study, which was the ATCC 14917. It would be important for the authors to cite a reference as to why they used this particular reference strain, since that there another strain of L. plantarum from ATCC, for example ATCC8014

2.- In the same section of matherial and methods, I suggest the authors standardize the labeling as ATCC with its space 14917 and not ATCC14917. page line #173

3.- In the Materials and Methods section I suggest standarize the labeling or description on page 4 line #156 of the 1% antibiotic-antimycotic (Anti-anti, Gibco, Waltham, MA, USA) than on the page 5 line #204 appears incomplete as Anti-anti. On this same page in line #209 it would be important to include the brand and specifications of the TNF-alpha cytokine that used, only the concentration is mentionated

Round 2

Reviewer 1 Report

The paper was revised to reflect the reviewer's comments.